# Impact of duration and magnitude of raised intracranial pressure on outcome after severe traumatic brain injury: A CENTER-TBI high-resolution group study

Cecilia Al Åkerlund[1,2☯‡]*, Joseph Donnelly[3‡], Frederick A. Zeiler[4,5,6,7,8☯], Raimund Helbok[9☯], Anders Holst[2☯‡], Manuel Cabeleira[3☯], Fabian Güiza[10☯], Geert Meyfroidt[10☯], Marek Czosnyka[3,11☯], Peter Smielewski[3☯], Nino Stocchetti[12☯], Ari Ercole[8☯‡], David W. Nelson[1☯‡], the CENTER-TBI High Resolution ICU Sub-Study Participants and Investigators[¶]

1 Department of Physiology and Pharmacology, Section of Perioperative Medicine and Intensive Care, Karolinska Institutet, Stockholm, Sweden, 2 School of Computer Science and Communication, KTH Royal Institute of Technology, Stockholm, Sweden, 3 Clinical Neuroscience, University of Cambridge, Cambridge, United Kingdom, 4 Section of Neurosurgery, Department of Surgery, Rady Faculty of Health Sciences, University of Manitoba, Winnipeg, Manitoba, Canada, 5 Department of Anatomy and Cell Science, Rady Faculty of Health Sciences, University of Manitoba, Winnipeg, Manitoba, Canada, 6 Biomedical Engineering, Faculty of Engineering, University of Manitoba, Winnipeg, Manitoba, Canada, 7 Centre on Aging, University of Manitoba, Winnipeg, Manitoba, Canada, 8 Division of Anaesthesia, University of Cambridge, Cambridge, United Kingdom, 9 Department of Neurology, Neurological Intensive Care Unit, Medical University of Innsbruck, Innsbruck, Austria, 10 Department and Laboratory of Intensive Care Medicine, University Hospitals Leuven and KU Leuven, Leuven, Belgium, 11 Institute of Electronic Systems, Warsaw University of Technolology, Warszawa, Poland, 12 Department of Pathophysiology and Transplants, University of Milan, and Neuroscience Intensive Care Unit, Fondazione IRCCS Cà Granda Ospedale Maggiore Policlinico, Milan, Italy

☯ These authors contributed equally to this work.
‡ These authors also contributed to the analysis of the data.
¶ A full list of the HR ICU Sub-Study Participants and Investigators is listed in the Acknowledgments.
* cecilia.akerlund@sll.se

**Citation:** Åkerlund CAI, Donnelly J, Zeiler FA, Helbok R, Holst A, Cabeleira M, et al. (2020) Impact of duration and magnitude of raised intracranial pressure on outcome after severe traumatic brain injury: A CENTER-TBI high-resolution group study. PLoS ONE 15(12): e0243427. https://doi.org/10.1371/journal.pone.0243427

**Data Availability Statement:** CENTER-TBI encourages data sharing, and there is a data

## Abstract

Magnitude of intracranial pressure (ICP) elevations and their duration have been associated with worse outcomes in patients with traumatic brain injuries (TBI), however published thresholds for injury vary and uncertainty about these levels has received relatively little attention. In this study, we have analyzed high-resolution ICP monitoring data in 227 adult patients in the CENTER-TBI dataset. Our aim was to identify thresholds of ICP intensity and duration associated with worse outcome, and to evaluate the uncertainty in any such thresholds. We present ICP intensity and duration plots to visualize the relationship between ICP events and outcome. We also introduced a novel bootstrap technique to evaluate uncertainty of the equipoise line. We found that an intensity threshold of 18 ± 4 mmHg (2 standard deviations) was associated with worse outcomes in this cohort. In contrast, the uncertainty in what duration is associated with harm was larger, and safe durations were found to be population dependent. The pressure and time dose (PTD) was also calculated as area under the curve above thresholds of ICP. A relationship between PTD and mortality could be established, as well as for unfavourable outcome. This relationship

sharing statement published: https://center-tbi.eu/data/sharing. Ethical and potential legal restrictions apply due to concerns regarding potential accidental re-identification of patients from such a complex and comprehensive dataset. Data will be available to researchers who provide a study proposal that is approved by the management committee to achieve the aims in the approved proposal. Proposals can be submitted online at https://www.center-tbi.eu/data. A request to access data to control analysis of a published article will be addressed in the same manner. A data access agreement is required and all access must comply with regulatory restrictions imposed on the original study.

**Funding:** Data used in preparation of this manuscript were obtained in the context of CENTER-TBI, a large collaborative project with the support of the European Union 7th Framework program (EC grant 602150). Additional funding was obtained from the Hannelore Kohl Stiftung (Germany), from OneMind (USA) and from Integra LifeSciences Corporation (USA). FAZ receives research support from the Manitoba Public Insurance (MPI) Neuroscience/TBI Research Endowment, the Health Sciences Centre Foundation Winnipeg, the United States National Institutes of Health (NIH) through the National Institute of Neurological Disorders and Stroke (NINDS), the Canadian Institutes for Health Research (CIHR), the Canadian Foundation for Innovation (CFI), the University of Manitoba Centre on Aging, the University of Manitoba VPRI Research Investment Fund (RIF), and the University of Manitoba Rudy Falk Clinician-Scientist Professorship. DN has also been funded by the Regional Research Agreement (ALF) with Stockholm City Council. Integra LifeSciences provided financial support in respect to data curation. The funders had no role in study design, data collection and analysis, decision to publish, or preparation of the manuscript.

**Competing interests:** The authors have read the journal's policy and have the following competing interests: ICM+ is a software created in Brain Physics Laboratory and licensed by Cambridge Enterprise Ltd, UK. PS and MC have financial interest in a fraction of licensing fee. Integra LifeSciences provided financial support in respect to data curation. This does not alter our adherence to PLOS ONE policies on sharing data and materials. There are no other patents, products in development or marketed products associated with this research to declare.

remained valid for mortality but not unfavourable outcome after adjusting for IMPACT core variables and maximum therapy intensity level. Importantly, during periods of impaired autoregulation (defined as pressure reactivity index (PRx)>0.3) ICP events were associated with worse outcomes for nearly all durations and ICP levels in this cohort and there was a stronger relationship between outcome and PTD. Whilst caution should be exercised in ascribing causation in observational analyses, these results suggest intracranial hypertension is poorly tolerated in the presence of impaired autoregulation. ICP level guidelines may need to be revised in the future taking into account cerebrovascular autoregulation status considered jointly with ICP levels.

## Introduction

Traumatic brain injury (TBI) is a major cause of worldwide mortality and morbidity [1]. A key goal of the neurointensive care of the most severely injured patients is to minimize secondary injury through interventions based on the close monitoring of intracranial and systemic physiology.

One of the most important physiological parameters in modern neurocritical care is intracranial pressure (ICP). Supported by a study on the TBI database in Cambridge, UK [2] the Brain Trauma Foundation (BTF) guidelines state that ICPs above 22 mmHg should be treated as this is associated with increased mortality [3], and in Europe there is a general consensus that ICP levels above 20 mmHg should be actively managed [4]. Despite this general consensus some studies have shed doubt on the efficacy of ICP monitoring itself [5–8] and question the validity of treating such fixed values. Indeed, without effective management strategies, monitoring by itself cannot improve outcome and heterogeneous treatment strategies may contribute as to explain a lack of established efficacy of monitoring. In particular absolute 'safe' levels of ICP have not conclusively been shown, however some attempts have been noted [9]. Additionally, there is no general consensus on what durations of increased ICP levels might be tolerated before harm is caused.

Automatic recording of physiological parameters has been shown to have advantages over manual detection of secondary insults in brain injuries [10, 11]. Continuous recording has made it possible to study the time and pressure dose of ICP in more detail. There has been increasing interest in the impact of the duration of elevated intracranial pressure, both in TBI, and in patients with other than acute brain syndromes. In two single-center studies [12, 13], with 93 and 60 TBI patients respectively, an association was found between an increased pressure-time dose of ICP and poor outcome at 6 months post injury. Similar results have been observed in a cohort of patients with spontaneous subarachnoid haemorrhage, although the underlying pathophysiology is likely to be different from TBI [14].

An important contribution to understanding the impact of insult duration on outcome was the insult intensity / duration plots described by Güiza et al [15] which correlated the number of events above increasing thresholds of pressure and time with outcome, visualizing the results on a colour-coded grid. The intensity/duration plot has shed important light on the relation between ICP events and their duration and outcome. Donnelly et al [16] produced a similar plot albeit with different cut-offs and using data from another cohort of TBI patients. The difference in results between the previous studies implies that ICP tolerability levels might not be universal, but cohort dependent and that the results are associated with some degree of uncertainty. As this uncertainty has not yet been investigated, and these types of plots may be

widely used to identify perceived safe levels and durations of raised ICP, it is essential to investigate and establish the certainty of these plots. The aims of this study are thus to investigate the impact of ICP intensity and duration on outcome in the large multi-center cohort in the CENTER-TBI study [17, 18], to examine the impact of cerebrovascular autoregulation status on ICP tolerability and to quantify the certainty/uncertainty of identifiable ICP injury thresholds.

## Materials and methods

High-frequency ICP (up to 500 Hz) and arterial blood pressure signals were recorded in 273 patients from 20 different sites participating in the European multi-center study CENTER-TBI, using the software ICM+ (Cambridge Enterprise Ltd, University of Cambridge, UK, versions 8.4.4.4 to 8.5.5.1), or a combination of ICM+ and CNS Monitor (Moberg Research Inc, Ambler, PA, USA), between January 2015 and March 2018. Pressure reactivity Index (PRx), the moving Pearson correlation between ICP and arterial blood pressure, was calculated using standard methodology in ICM+ [19, 20]. Data for the CENTER-TBI study was collected through the Quesgen e-CRF (Quesgen Systems Inc, USA), hosted on the INCF platform and extracted via the INCF Neurobot tool (INCF, Sweden). Version 2.1 of the CENTER-TBI dataset was used in this manuscript.

All patients met the general inclusion criteria for CENTER-TBI (Clinical diagnosis of TBI, clinical indication for CT scan and presentation within 24 hours of injury) and were admitted directly from the ER to the ICU [17]. This study was approved by the CENTER-TBI management committee. The CENTER-TBI study was conducted in accordance with all relevant laws of the European Union if directly applicable or of direct effect and all relevant laws of the country where the Recruiting sites were located, including but not limited to, the relevant privacy and data protection laws and regulations (the "Privacy Law"), the relevant laws and regulations on the use of human materials, and all relevant guidance relating to clinical studies from time to time in force including, but not limited to, the ICH Harmonised Tripartite Guideline for Good Clinical Practice (CPMP/ICH/135/95) ("ICH GCP") and the World Medical Association Declaration of Helsinki. Written or oral Informed Consent by the patients or next of kin was obtained, accordingly to the local legislations, for all patients recruited in the Core Dataset of CENTER-TBI and documented in the electronic case report form. In case of oral consent, a written confirmation was requested.

Ethical approval was obtained for each recruiting site. The list of sites, Ethical Committees, approval numbers and approval dates are available online [21] and ethical approval numbers for sites having recruited patients to the high-resolution sub-study of CENTER-TBI is listed in S1 Appendix.

### Data preparation

One-minute averages of ICP data were calculated from 10-second summaries. Data from patients with ventriculostomies was included: External ventricular drains (EVD) were confirmed to have been closed throughout the monitoring period by manual inspection of the ICP waveforms. Data from the day of trauma through day 7 were used for the calculation of ICP burden, based on previous results that mean ICP differs between survivors and non-survivors only the first 7 days post injury [22].

The Glasgow Outcome Scale Extended (GOS-E) 6 months post injury was used as outcome measure, where 1 indicates death and 8 good recovery without disability. If GOS-E scores at 6 months were missing, a derived GOS-E score was used. A multi-state model created centrally in CENTER-TBI was used if at least one GOS-E value was present outside the pre-specified

time window for 6 months. If GOS-E score was missing and could not be imputed, the patient was removed from the final analysis.

Forty-six patients were excluded due to monitoring time shorter than one day ($n = 8$), missing GOS-E score at 6 months ($n = 30$) or missing baseline data ($n = 8$), leaving 227 patients for the final analysis.

### Correlations between number of events above thresholds of pressure and time and outcome

The correlation between number of insults above thresholds of ICP and duration was calculated using the insult intensity/duration plot method described previously [15]: The correlations are presented in a grid where each pixel is represented by a colour (blue = positive i.e. better outcome, red = negative, i.e. worse outcome). It corresponds to the Pearson correlation coefficient between mean number of events and GOS-E score for each position on the plot. Each position on the plot represents events above the given ICP level and of a duration. Thresholds of ICP between 10 and 40 mmHg and duration from 5 to 360 minutes were used generating a grid of 11,036 pixels.

PRx, the moving Pearson correlation between ICP and arterial blood pressure, was provided in the measurement files. By averaging PRx for each event, all events were classified into either impaired (PRx > +0.3) or intact (PRx <= 0.3) autoregulation. The cut-off of PRx +0.3 was chosen as threshold, as it previously has been suggested to be associated with worse outcome [2, 23, 24]. Correlations of events with either impaired or intact autoregulation were represented in separate grids. Additionally, we expanded this method as to investigate uncertainty and variability of the results with variations in patient cohorts. This was done using bootstrapping with replacement, generating 1,000 different cohorts. This is a technique where new cohorts are generated by randomly selecting 227 patients. The replacement condition implies that any patient can occur more than once in each sampled cohort. By averaging and calculating standard deviations of the correlations at each grid point, the stability and uncertainty and of the equipoise lines were investigated. By averaging the bootstrapped correlations, a mean transition line was created. Standard deviations of the correlations at each grid point were also calculated, and lines representing correlations +-2 standard deviations from the mean transition line were created.

All analyses were performed using R version 1.1.453 [25].

### Pressure and time dose of ICP (PTD)

In addition to the intensity / duration plots, we investigated the pressure and time dose of ICP (PTD) as a simple alternative measure of insult severity. The PTD was calculated as the area under the curve above thresholds of ICP from 0 to 40 mmHg, as illustrated in Fig 1. Mean doses were calculated for patients with unfavourable/favourable outcome as well as for patients who were dead or alive within 6 months post injury. PTD for intact and impaired autoregulation respectively was also calculated.

GOS-E score 5 to 8 was defined as favourable and 1 to 4 as unfavourable outcome. Comparisons of distributions of PTD between groups were performed using the non-parametric Kolmogorov-Smirnov test. A threshold of 0.05 was chosen for statistical significance.

To investigate the relationship between ICP event-burden and PTD towards outcome, we performed multivariable regression analyses, adjusting for known covariates including the IMPACT core variables age, GCS motor score and pupil reactivity [26, 27] and maximum daily therapy intensity level (TIL) score.

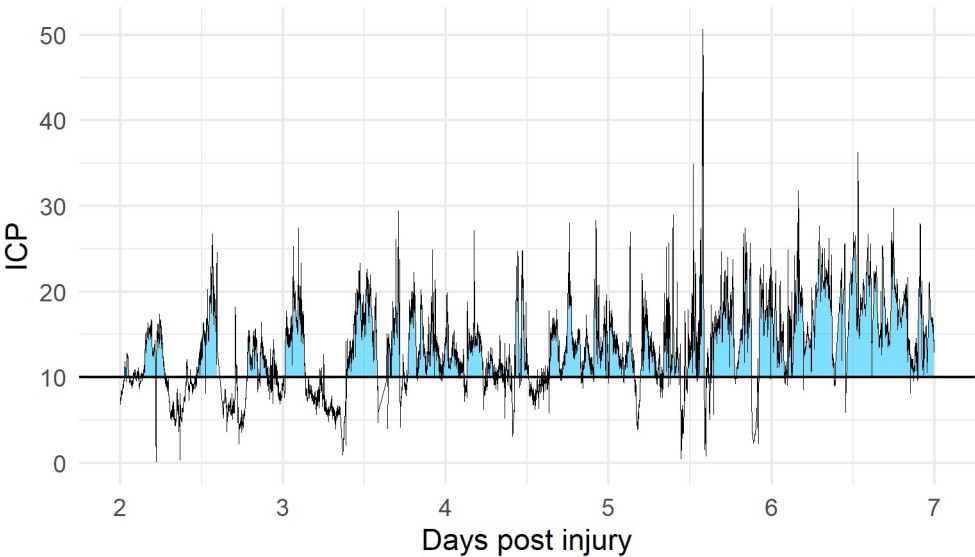

**Fig 1. Calculation of AUC of ICP over time.** An example of how ICP dose as pressure times time dose is represented from a representative patient. The blue-coloured area is the AUC (i.e. the ICP dose) above threshold ICP = 10 mmHg and represents the PTD10.

## Time spent above transition line

To investigate the impact of co-variates on the time spent in areas of the grid with correlation to worse outcome, the time above the transition line was calculated. This was done by first determining the duration of all intensity thresholds at the transition line. For each patient and intensity threshold, the durations of events above this threshold was calculated. If the duration was longer than the duration threshold for that intensity threshold, all ICP values in that episode was regarded to be above the transition line. The duration of all ICP values above the transition line was summarized and divided by total monitoring time.

## Results

### Patient characteristics

227 patients with high-resolution ICP measurement for more than one day, over 18 years old and with 6 month GOS-E were included in the final analysis. As presented in Table 1, our cohort consisted of 79% males with a median age of 51 years (IQR 32–64), and 50.2% had no comorbidities at time of injury (ASA class 1), indicating a fairly healthy population before injury. With a median pre-ICU Glasgow Coma Score (GCS) of 6 (IQR 3–10), the cohort can be classified as moderate to severe TBI. 53 patients (23%) underwent a decompressive craniectomy. The median GOS-E at 6 months post injury was 4 (IQR 2–5), Fig 2.

### Correlations between number of insults and outcome

The correlation between number of events against ICP and duration for the high-resolution cohort is presented in Fig 3A. A black transition curve divides the surface into two areas: A small blue area in the bottom left corner where number of events more frequently occur in patients with better outcome and a large red-orange area where number of events are associated with worse outcome. The transition curve represents a no correlation region between number of events and outcome.

**Table 1. Characteristics of the cohort.**

| Demographic characteristics | |
|---|---|
| Age (years) | 51 (32–64) |
| Sex | |
| Female | 48 (21.1) |
| Male | 179 (78.9) |
| **Pre-injury health status** | |
| ASA-PS classification | |
| 1 | 109 (50.2) |
| 2 | 79 (36.4) |
| 3 | 28 (12.9) |
| 4 | 1 (0.5) |
| **Cause of injury and injury severity** | |
| Cause of injury | |
| Road traffic incident | 94 (41.4) |
| Incidental fall | 83 (36.6) |
| Other non-intentional injury | 9 (4) |
| Violence/Assault | 18 (7.9) |
| Suicide attempt | 3 (1.3) |
| Unknown | 11 (4.8) |
| Other | 9 (4) |
| ISS | |
| Total | 34 (25–43) |
| Highest Extracranial | 9 (0–16) |
| Highest Head/Brain/Cervical | 25 (25–25) |
| **Clinical presentation** | |
| GCS (best pre-hospital) | |
| Motor score | 4 (1–5) |
| Total score | 6 (3–10) |
| Pupillary reactivity (at baseline) | |
| Both reacting | 152 (72.7) |
| One reacting | 16 (7.7) |
| None reacting | 41 (19.6) |
| Hypoxia (pre-ICU admission) | |
| No | 160 (81.6) |
| Definite | 19 (9.7) |
| Suspect | 17 (8.7) |
| Hypotension (pre-ICU admission) | |
| No | 171 (86.8) |
| Definite | 18 (9.1) |
| Suspect | 8 (4.1) |
| **CT characteristics** | |
| Rotterdam CT Score | 4 (3–5) |
| Contusion | 151 (74.4) |
| Cisternal compression | 94 (46.3) |
| Skull fracture | 129 (63.5) |
| Midline shift > 5 mm | 67 (33) |
| Mass lesions > 25 ml | 106 (52.2) |
| tSAH | 175 (86.2) |

(*Continued*)

**Table 1.** (Continued)

| | |
|---|---|
| EDH | 45 (22.2) |
| aSDH | 127 (62.6) |
| cSDH | 27 (13.3) |
| IVH | 77 (37.9) |
| **Other characteristics** | |
| Hypoxia (during hospital stay) | |
| No | 156 (69.3) |
| Single episode, short duration | 56 (24.9) |
| Multiple episodes or prolonged duration | 13 (5.8) |
| Hypotension (during hospital stay) | |
| No | 136 (60.4) |
| Single episode, short duration | 61 (27.1) |
| Multiple episodes or prolonged duration | 28 (12.4) |
| Type of ICP device | |
| Ventricular | 18 (7.9) |
| Ventricular + inbuilt sensor | 5 (2.2) |
| Parenchymal | 191 (84.1) |
| Other | 13 (5.7) |
| Decompressive craniectomy | 53 (23.3) |
| Length of stay, days | 23.73 (11.9–46.7) |
| Length of stay in ICU, days | 13.52 (8.7–20.1) |
| Monitoring time, days | 5.18 (3.7–7.2) |
| Mean ICP, mmHg | 12.62 (9.4–15.4) |
| Mean body temperature, ˚C | 37.07 (36.7–37.4) |
| Mean CPP, mmHg | 70.99 (65.3–77.1) |
| Sodium day 2 post injury (mmol/L) | 142 (139–146) |
| **Outcome** | |
| GOS-E at 6 months | |
| 1 | 54 (23.8) |
| 2 | 10 (4.4) |
| 3 | 47 (20.7) |
| 4 | 24 (10.6) |
| 5 | 36 (15.9) |
| 6 | 29 (12.8) |
| 7 | 12 (5.3) |
| 8 | 15 (6.6) |

Data are median (IQR) or n (%). ASA-PS classification: American society of anesthesiologists physical status classification, ISS: Injury Severity Score, GCS: Glasgow Coma Scale, ICU: Intensive care unit, Rotterdam CT Score: a score describing the severity of findings on a CT scan, CT: Computed tomography, tSAH: Traumatic subarachnoidal haemorrhage, EDH: Epidural hematoma, aSDH: Acute subdural hematoma, cSDH: Chronic subdural hematoma, IVH: Intra-ventricular haemorrhage, ICP: Intracranial pressure, CPP: Cerebral perfusion pressure, GOS-E: Glasgow Outcome Scale extended.

We investigated the stability of the results by applying bootstrapping with replacement to create 1,000 different populations of 227 patients (same sample size as our cohort) to give the population dependent variability of the transition line (corresponding to correlation coefficient 0 towards GOS-E). Ten randomly selected bootstrapped correlation plots are presented

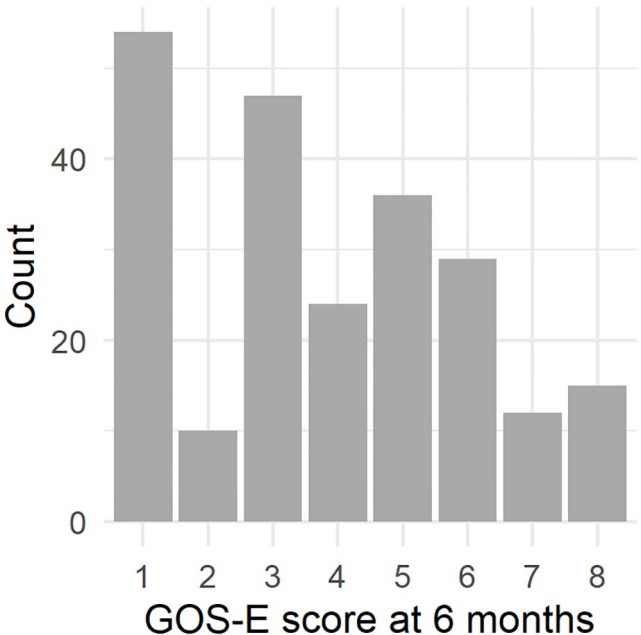

**Fig 2. Distribution of GOS-E score at 6 months.**

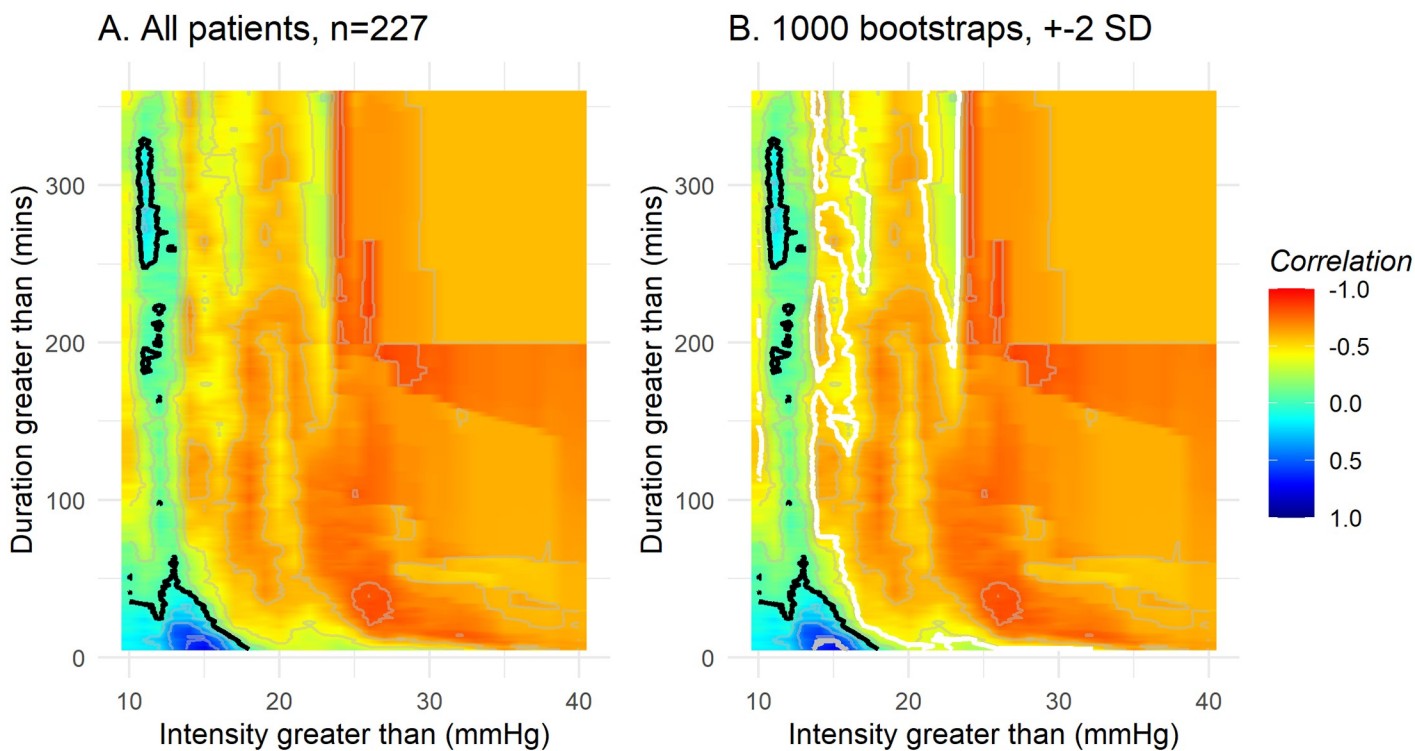

**Fig 3. Correlation between number of events above thresholds of intracranial pressure and durations, and outcome (GOS-E score).** Red indicates that ICP events are correlated to worse outcome at that specific ICP level and event duration on the map. **A.** The black line represents the transition line, where there is no correlation between number of events above threshold and outcome. **B.** The black line represents the mean transition line of 1000 bootstraps. The white lines represent the mean transition line +2 SD, while the grey line represents the mean transition line -2 SD. Above, and to the right, of the white line, there is a high degree of statistical certainty of events being associated with worse outcome, whereas below the grey line, the statistical certainty is high that events are not associated with harm.

in S1 Fig to illustrate how the results are affected by different cohort constitutions. Mean correlations, with the mean transition line in black (worse vs. better outcome), plus/minus two standard deviations (white), are presented in Fig 3B.

To investigate the impact of cerebral autoregulation status on tolerability of ICP events, all events were stratified according to either intact (mean PRx <= 0.3) or impaired (mean PRx > 0.3) autoregulation, Fig 4A and 4B. All patients had, to different extents, both events with intact and impaired autoregulation, Fig 5, and 24.9% of the total monitoring time had a mean PRx > 0.3, indicating impaired autoregulation. In case of impaired autoregulation (Fig 4B), no threshold for tolerable ICP intensities and durations could be found.

In the univariable regression analysis, the time spent above the transition line was a statistically significant predictor of both unfavourable outcome and 6-month mortality, OR = 2.24 (95% CI 1.02–4.99, $p = 0.046$) and OR = 4.18 (95% CI 1.64–11.16, $p = 0.003$). When adjusted for the IMPACT core variables and maximum daily TIL, time above the transition line remained statistically significantly associated with mortality OR = 3.56 (95% CI 1.14–11.74, $p = 0.032$), but not with unfavourable outcome OR = 1.37 (95% CI 0.51–3.76, $p = 0.533$). A full summary of the regression is presented in Table C in S2 Appendix.

## Pressure and time dose of ICP

The mean PTD above thresholds of 0 to 40 mmHg are presented for each category of GOS-E in Fig 6. Patients with unfavourable outcome had a significantly higher mean PTD above 20

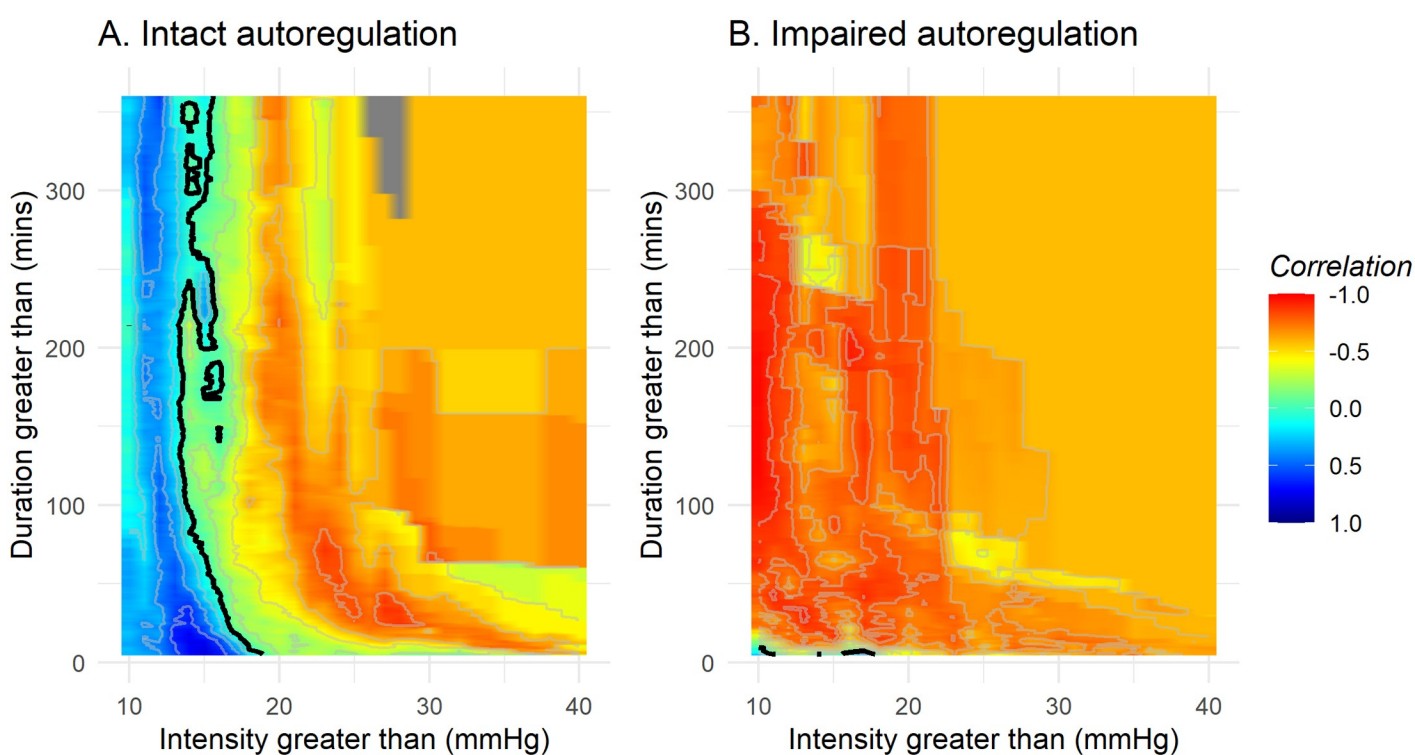

**Fig 4. Correlation between number of events above thresholds of intracranial pressure intensity and duration and outcome, stratified by cerebral autoregulatory status.** Orange / red areas indicates areas where ICP levels and event durations are associated with worse outcomes. The transition line, i.e. where there is no correlation between number of events and outcome, is drawn in black. All patients contribute some data to both plots, the degree however depending on the extent of their intact vs. impaired autoregulation. A) Intact autoregulation (mean PRx <= 0.3), B) Impaired autoregulation (mean PRx > 0.3).

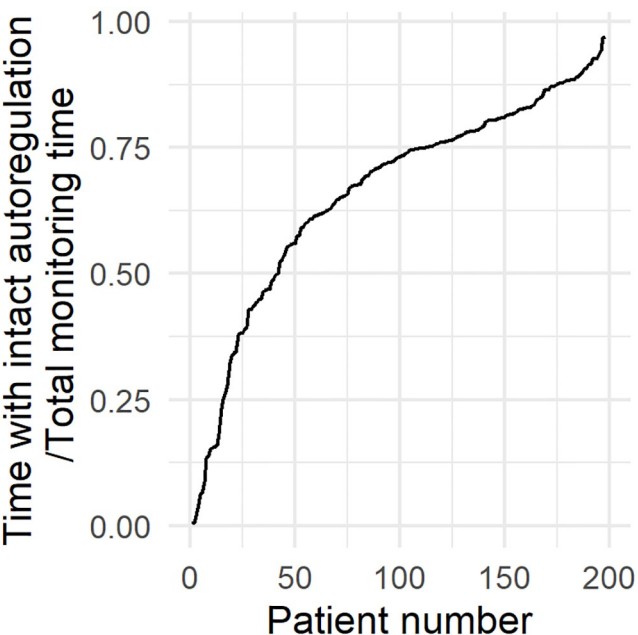

**Fig 5. Fraction of monitoring time with intact autoregulation, per patient.** All patients had both episodes of impaired and intact autoregulation, but to different extents.

and 25 mmHg compared to patients with favourable outcome, Fig 7A and Table 2, with PTD20 being 232.9 (± 750.8) vs 35.1 (± 64.5) mmHg·h respectively ($p$ = 0.014).

On average, patients who died within 6 months post injury had a statistically significantly higher PTD above all ICP thresholds of 10 mmHg and above compared to survivors, Fig 7B and Table 2. The mean PTD above 20 mmHg was 493.4 (± 1125.6) vs 43.8 (± 84.7) mmHg·h ($p$ = 0.004).

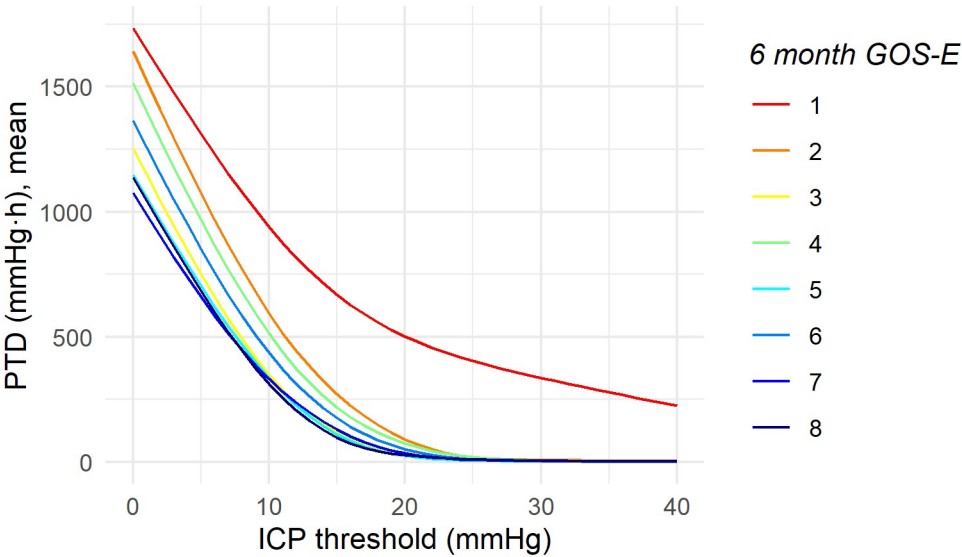

**Fig 6. Total pressure and time dose (PTD) above ICP thresholds stratified by outcome at 6 months.** The group mean PTD was higher for patients who had died within 6 months post injury, while the mean doses were similar for GOS-E 3 to 8 outcomes.

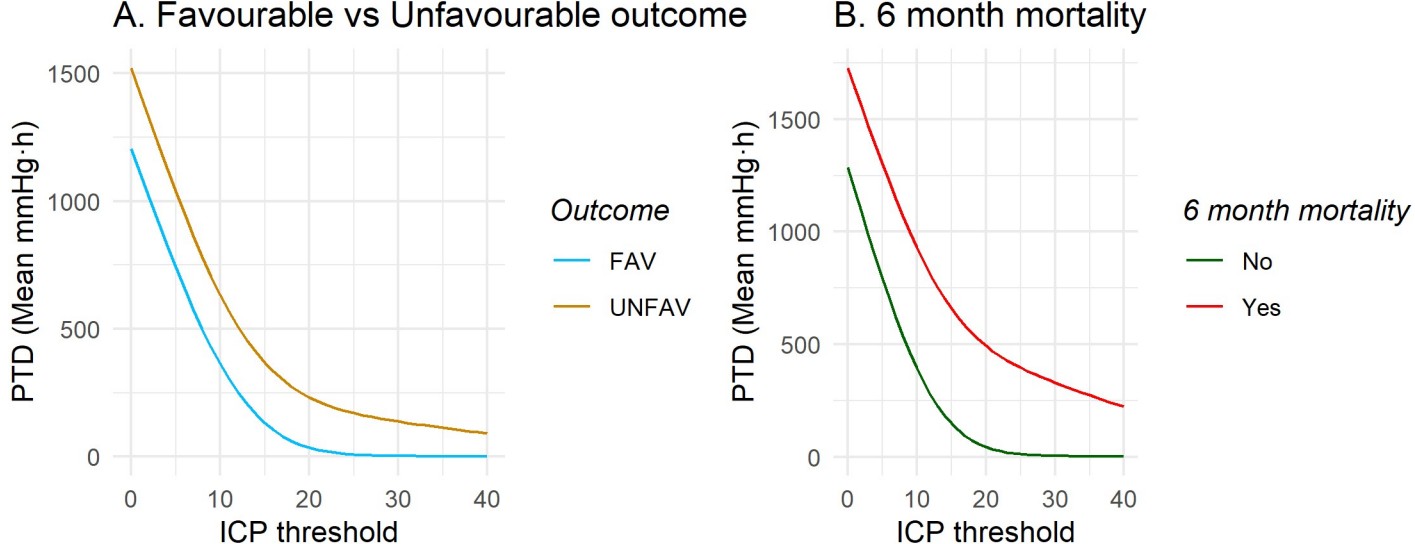

**Fig 7. Group mean PTD (mmHg·h) above thresholds of ICP, median for A) favourable vs unfavourable outcome, B) dead vs alive at 6 months post injury.**

PTD was also calculated separately for periods with intact (PTDintact) and impaired (PTDimpaired) autoregulation, Fig 8, Tables 3 and 4. There was no significant difference in mean PTD between favourable and unfavourable outcome at any ICP threshold when stratified by intact or impaired autoregulation, Fig 8A and Table 3.

Additionally, autoregulation stratified mortality cohorts showed similar patterns to that of unstratified mortality but with a generally stronger association towards outcome. PTDintact and PTDimpaired stratified cohorts were significant related to mortality above all thresholds of ICP, Fig 8C and 8D and Table 4. The mean PTD of intact autoregulation above 20 mmHg was 66.7 (± 150.6) vs 25.0 (± 64.5) mmHg·h for non-survivors and survivors, respectively ($p = 0.0037$), and mean PTDimpaired above 20 mmHg was 414.9 (± 1035.6) and 12.8 (± 20.1), respectively ($p<0.001$).

When adjusted for the IMPACT core variables (age, GCS motor score at baseline and pupil reactivity) and maximum daily TIL, PTD was not an independent predictor of favourable outcome, OR = 1.0 (95% CI 0.99–1.00, $p = 0.390$), but still a significant predictor for 6 month mortality, OR = 1.0 (95% CI 1.00–1.01, $p = 0.012$), Table D in S2 Appendix. Neither PTDintact nor PTDimpaired were retained as significant predictors of 6-month mortality in a multivariable regression model, OR = 1.00 (95% CI 0.99–1.01, $p = 0.238$) and 1.02 (1.00–1.02, $p = 0.236$), respectively.

**Table 2. Group mean PTD (mmHg·h) above thresholds of ICP.**

| PTD | Favourable outcome | Unfavourable outcome | p | Alive at 6 months | Dead at 6 months | p |
|---|---|---|---|---|---|---|
| 0 | 1205.4 (±632.4) | 1519.9 (±1118.9) | 0.160 | 1285.8 (±649.8) | 1725.9 (±1547.3) | 0.080 |
| 10 | 363.7 (±335) | 633.3 (±956.6) | 0.236 | 394.0 (±372.1) | 930.8 (±1368.3) | 0.004 |
| 15 | 132 (±175.6) | 367.8 (±853.7) | 0.173 | 148.8 (±210.1) | 658.2 (±1255.7) | 0.003 |
| 20 | 35.1 (±64.5) | 232.9 (±750.8) | 0.014 | 43.8 (±84.7) | 493.4 (±1125.6) | 0.001 |
| 25 | 8.7 (±16.3) | 170.6 (±659.4) | 0.017 | 11.9 (±24.7) | 396.3 (±994.9) | 0.003 |
| 30 | 3.2 (±6.2) | 137.9 (±572.6) | 0.071 | 4.5 (±10.6) | 329.9 (±866.1) | 0.003 |

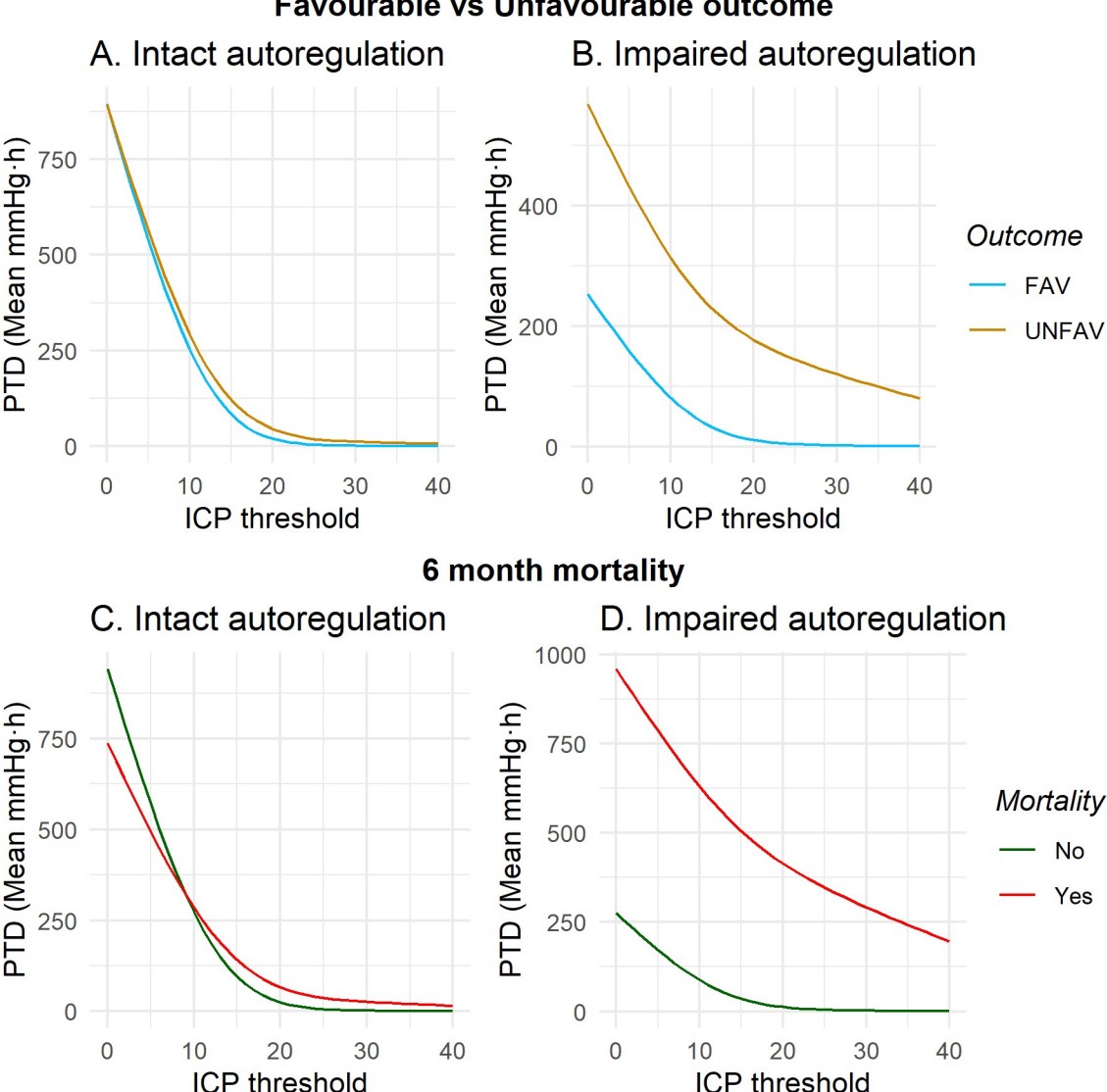

**Fig 8. A) Intact vs B) impaired autoregulation, mean PTD for favourable and unfavourable outcome, C) Intact vs D) Impaired autoregulation for 6 month mortality.**

## Discussion

In this study we investigate the relationship between time-dependent ICP insults and outcome. We confirm findings of the Insult intensity/duration plot methodology, albeit finding lower acceptable ICP levels, in a new multicentre cohort and also investigate simpler pressure-time-dose measures. Importantly, we also introduce a novel bootstrap methodology to assess the certainty/uncertainty of the transition line of the correlation plot, above which there is an increased correlation between number of events and worse outcome. We believe this is a necessary extension to the insult intensity plots in order to interpret them with confidence. Additionally, we investigate relations of potential ICP vulnerability during periods of intact and impaired autoregulation suggesting that safe ICP levels may vary depending on autoregulatory status.

**Table 3. Group mean PTD (mmHg·h) of intact or impaired autoregulation above thresholds of ICP, favourable vs unfavourable outcome.**

| | Intact autoregulation | | | Impaired autoregulation | | |
|---|---|---|---|---|---|---|
| PTD | Favorable Outcome | Unfavorable outcome | p | Favorable Outcome | Unfavorable outcome | p |
| 0 | 893.4 (±531) | 892.5 (±590.3) | 0.870 | 252.6 (±171.9) | 568.2 (±1009.9) | 0.030 |
| 10 | 253.4 (±264) | 291.9 (±328.7) | 0.900 | 80.7 (±74.1) | 312.9 (±847.5) | 0.077 |
| 15 | 84 (±132.2) | 121.9 (±203.3) | 0.788 | 32.1 (±36.3) | 228.7 (±766.8) | 0.073 |
| 20 | 19.8 (±50.4) | 45.6 (±113.9) | 0.222 | 10.8 (±14.4) | 177.2 (±684.2) | 0.031 |
| 25 | 4.6 (±12.4) | 19.4 (±79.5) | 0.090 | 4 (±6.5) | 144.3 (±603.2) | 0.068 |
| 30 | 1.5 (±3.6) | 12.1 (±65.6) | 0.248 | 1.7 (±3.5) | 120 (±524.8) | 0.081 |

**Table 4. Group mean PTD (mmHg·h) of intact or impaired autoregulation above thresholds of ICP, Dead or alive at 6 months, intact and impaired autoregulation.**

| | Intact autoregulation | | | Impaired autoregulation | | |
|---|---|---|---|---|---|---|
| PTD | Alive at 6 months | Dead at 6 months | p | Alive at 6 months | Dead at 6 months | p |
| 0 | 941 (±553.1) | 738.1 (±581.3) | 0.020 | 275.1 (±189.5) | 959.2 (±1491.3) | < 0.001 |
| 10 | 272.8 (±297.5) | 285.9 (±325) | 0.760 | 88 (±82.4) | 630.4 (±1269) | 0.002 |
| 15 | 95.2 (±162.6) | 141.5 (±218.4) | 0.327 | 35.8 (±44) | 505.3 (±1155.2) | 0.001 |
| 20 | 25 (±64.5) | 66.7 (±150.6) | 0.037 | 12.8 (±20.1) | 414.9 (±1035.6) | < 0.001 |
| 25 | 5.9 (±16.8) | 36.8 (±121) | 0.021 | 4.9 (±9.6) | 347.4 (±915.7) | 0.001 |
| 30 | 1.9 (±5.6) | 26.3 (±101.8) | 0.027 | 2.2 (±5.7) | 292 (±798.5) | 0.006 |

The overall pattern concerning association of ICP events and worse outcome in our study is similar to previously published results: However, we find lower limits for acceptable ICPs. In contrast to ICP thresholds, there is a higher degree of uncertainty in what duration of insult is associated with harm; the length of ICP events that are associated with worse outcome were more variable and population dependent. As indicated in Fig 3B, ICP levels above 18 ±4 mmHg for five minutes or longer are associated with worse outcome whereas another threshold at 13 mmHg with a wide time uncertainty is seen.

Above the +2 SD line there is a strong association between number of events at all levels of intensity and duration of ICP and worse outcome, and the time spent in this zone is strongly correlated both to 6-month mortality and unfavourable outcome. It can also be concluded that it is fairly certain that events to the right of the white line (mean transition line +2 SD) are associated with worse outcome. This corresponds to ICP events above 22 mmHg longer than 5 minutes and above 16 mmHg for longer than 60 minutes. As a comparison, previously suggested cut-offs from Güiza et al are 35 mmHg for 5 minutes or 20 mmHg for 37 minutes [15] and from Donnelly et al an ICP of 20 mmHg for longer than 13 minutes [16]. However, in both cases, correlations between 5-level GOS (rather than GOS-E as in this work) and ICP events were analysed. The first mentioned cohort was similar to ours with respect to age, admission GCS and cerebrovascular reactivity status, however the CENTER-TBI high-resolution cohort had a worse outcome at the group level. This may be attributable to numerous possible factors and could cause a general shift in our curves. To investigate if the difference in results were not due to using different outcome measures (GOS vs. GOS-E), we reproduced the same analysis with GOS as an outcome measure, with almost no difference in result (S3 Fig). It is important to mention that these results assume a linear relationship between number of events above thresholds and GOS-E score, which might be an inappropriate approximation and needs to be explored further. In summary, we find that although the pattern is similar between cohorts, absolute levels differ, supporting

our effort to investigate certainty/uncertainty on regions of the map before defining or suggesting generalized cut-offs.

As with any observational analysis, the question is whether our results represent causation (i.e. if reducing ICP to lower insult levels in a timely way could affect outcome) or simply association, and the methods used in this study cannot distinguish between the two. However, it is biologically plausible that short periods of higher levels of ICP could be causally related to outcome although perhaps identified thresholds for longer time periods might represent associations at a cohort level as uncertainty is higher along this direction of the map.

The association between ICP intensity / duration and outcome are likely to be affected by any treatment directed at lowering of ICP. An attempt to adjust for these factors was done by including a measure of therapy intensity level in a multivariable regression model of time in red-orange zone and its association on outcome. This analysis suggested that the time above the transition line is a statistically significant predictor for death but not for unfavourable outcome.

Decompressive craniectomy (DC) is an intervention most likely affecting intracranial pressure levels as well as tolerability and reactivity. In the analysis, the 53 patients in our cohort who underwent DC were included. Including these patients might be regarded a limitation, particularly when previous studies have been inconsistent in whether PRx is affected or not [28, 29]. However, it is biologically plausible that ICP elevations are harmful to the brain per se, no matter whether DC has been performed or not. A separate analysis of this group alone could not be performed due to limited sample size and a seemingly greater inter-individual variation of ICP tolerability. A sensitivity analysis, excluding these patients is presented in S2 Fig yielded qualitatively similar results.

Broadly speaking, although our results suggest a somewhat lower threshold of ICP elevations our results are not dissimilar to both the BTF recommendations (ICP target below 22 mmHg [8]) and European neurointensive care practice (ICP 20 mmHg [4]) especially if bearing in mind that the error of measurement for ICP measurement is of the order of 1.5 mmHg [30].

Importantly, we also demonstrate that ICP tolerability appears highly dependent on cerebrovascular reactivity, Fig 4, and there appears to be no threshold for tolerable ICP during periods of disrupted autoregulation. In the case of intact autoregulation, our results suggest that an ICP above 19 mmHg for 5 minutes or longer or 15 mmHg for 50 minutes or longer was strongly associated with worse outcome. The finding that ICP tolerability may be dependent on cerebral autoregulation status is in line with previous studies which have suggested that autoregulation status or individualized ICP thresholds derived from autoregulation status seem to better predict outcome than fixed ICP levels [9, 31]. A similar pattern was also found by Güiza [15]. Further, it is biologically plausible that in the absence of intact autoregulation, the brain is left vulnerable and unable to compensate for global and regional changes in cerebral perfusion over time. Attempts to determine individual baseline ICPs by correlating the PRx to ICP to identify individualized ICP thresholds has been done by both Lazaridis et al and Zeiler et al [9, 31], an approach that needs to be further investigated in future studies. Several recent publications have also pointed at cerebrovascular reactivity being more important than fixed ICP thresholds in limiting secondary injuries [32–34]. It has also been suggested that it is the cerebral perfusion pressure (CPP) rather than the ICP that represent the true burden of secondary insult. The purpose of this study, however, was to investigate the impact of ICP, and the impact of treatments to optimise care (e.g. targeting CPPopt [35]) still need to be explored.

Our study and previous studies indicate that avoiding ICP peaks above 20 mmHg appears justified in aggregate across all patients, however during periods of autoregulatory loss, no safe limit can be identified.

## The role of pressure times time dose

The PTD may be an additional, simple, measure of ICP tolerability and has been suggested as a predictor for mortality and unfavourable outcome by several authors, especially doses of ICP 20 mmHg and above. However, methodology and choice of threshold for PTD has been varied between studies, making comparison difficult [12–14, 36]. As seen in Fig 6, we also identify a relationship between higher PTD and worse outcome. A statistical difference was identified between a PTD above 20 mmHg and unfavourable outcome, and that of PTD above ICP 10 mmHg and mortality. This may not represent absolute levels of ICP but a general association that is merely stronger in relation to mortality vs. favourable/unfavourable outcome. However, an association that, when adjusted for IMPACT core variables and TIL, remained significant for mortality but not unfavourable outcome (S2 Appendix Table B). Care is required in the final interpretation of PTD cut-off levels as ICP vulnerability might be expected to change over time with changing pathophysiology, as well as the individual metabolic states of the brain, and a next step would be to investigate the temporal evolution of ICP and ICP vulnerability.

Our results differ from previously reported PTD, which may reflect differences in the details of the methods used in its calculation with time windows from 24 hours up to total monitoring time as well as different time resolutions of ICP measurements. There is no choice that is clearly superior as a description of the ICP secondary insult burden. Nevertheless, despite these discrepancies, a relationship between PTD and outcome appears robust across studies.

In summary, we identify a clear relationship between high doses of ICP and mortality but not for favourable/unfavourable outcome. However, we also identify clear correlations between number of events above thresholds of ICP intensity, even with short durations, and worse outcome. In aggregate this could suggest that peaks of ICP elevation might in themselves be harmful. We hypothesize, given the greater uncertainty in the time dimension, that short periods of raised ICP may be causal of injury and long periods of moderately high ICP may be association with injury severity. If so, ICP variability may also be related to outcome and worth further future investigation.

## Advantages, limitations and future directions

A major strength of this study is its multi-center design where more than 220 patients from 20 sites across Europe were included. This, in combination with the applied bootstrapping technique minimizes potential confounding effects of site-specific treatments of severe TBI.

Some limitations must be noted. It is important to stress that the relationship which we have established between ICP magnitude and duration and outcome is associative, and none of the methods used in this paper can conclusively establish a causative relation. We have also made the assumption of a linear relationship, and cannot exclude that a non-linear model would make a better fit. Treatment of patients with severe TBI is complex, and the impact of treatments on outcome is not yet fully understood. Although we have adjusted for treatments in multivariable regression models, we have not been able to show if they act as confounders or are causative in the relationship of ICP and outcome. Furthermore, ICP vulnerability might change over time, something we have not taken into consideration in this study.

It would be of great interest to further investigate aspects of the temporal evolution of ICP and ICP vulnerability as well as to better establish potential causality in relation to outcome. Our results lead us to hypothesize that short periods of ICP elevations may be causal of injury and more extended periods of moderate elevation may be more associated to TBI severity.

Future studies should focus on exploring the use of emerging techniques to evaluate causality with mathematical modelling and to investigate the impact of ICP variability on outcome.

## Conclusions

We have explored the relations of ICP towards outcome, employing several metrics of burden. We identify ICP limits and event durations associated with worse outcome, and importantly the uncertainty of such estimates. We find 18 mmHg to be the most probable safe ICP limit even for short durations. Given an uncertainty of ± 4 mmHg (± 2 SD), 22 mmHg can be identified as a limit that is with a high certainty related to worse outcome, and thus in concert with current BTF guidelines. However, it is lower than earlier event-duration plot studies. Although the adjusted ICP pressure time dose was strongly correlated to mortality, short periods of high ICP appear more confidently related to worse outcome than long periods of moderately high ICP leading us to hypothesize that the relation of burden towards outcome at lower ICP levels may be an association with injury severity, but shorter periods of elevated ICP may be more causative of injury. Additionally, we have found that ICP tolerability appears highly dependent on the cerebral autoregulation status where, in the case of impaired cerebrovascular reactivity, no safe ICP levels could be identified, suggesting that safe limits may need to be related to current autoregulatory status in the future.

## Supporting information

**S1 Appendix. List of ethical approvals for sites included in the high-resolution CEN-TER-TBI sub-study.**
(XLSX)

**S2 Appendix. Supplementary Tables.**
(DOCX)

**S1 Fig. Correlation between number of events above thresholds of intracranial pressure and duration and outcome (GOS-E score), ten illustrative bootstraps with replacement, sample size 209.** The black line represents the transition line, above which there is a correlation between more events and worse outcome. As seen, the shape and values of the transition line is dependent on the patient selection.
(TIF)

**S2 Fig. Correlation between number of events above thresholds of intracranial pressure and duration and outcome (GOS-E score).** A. All patients. B. All patients who has not undergone decompressive craniectomy. C. All patients with other monitors than extra-ventricular drain. **D.** All patients who has not undergone decompressive craniectomy and do not have an extra-ventricular drain.
(TIF)

**S3 Fig. Correlation between number of events above thresholds of intracranial pressure and duration and outcome (GOS score).**
(TIF)

**S4 Fig. Distribution of monitoring time in days, stratified by 6 month mortality.**
(TIF)

**S5 Fig. Distribution of mean PTD above thresholds of ICP 0 to 30 stratified by 6 month mortality status.**
(TIF)

## Acknowledgments

The authors would like to thank the CENTER-TBI High Resolution ICU Sub-Study Participants and Investigators for taking part in the collection of data:

Group leader: Professor Nino Stocchetti: nino.stocchetti@unimi.it

Audny Anke [1], Ronny Beer [2], Bo-Michael Bellander [3], Erta Beqiri [4], Andras Buki [5], Manuel Cabeleira [6], Marco Carbonara [7], Arturo Chieregato [4], Giuseppe Citerio [8, 9], Hans Clusmann [10], Endre Czeiter [11], Marek Czosnyka [6], Bart Depreitere [12], Ari Ercole [13], Shirin Frisvold [14], Raimund Helbok [2], Stefan Jankowski [15], Danile Kondziella [16], Lars-Owe Koskinen [17], Ana Kowark [18], David K. Menon [13], Geert Meyfroidt [19], Kirsten Moeller [20], David Nelson [3], Anna Piippo-Karjalainen [21], Andreea Radoi [22], Arminas Ragauskas [23], Rahul Raj [21], Jonathan Rhodes [24], Saulius Rocka [23], Rolf Rossaint [18], Juan Sahuquillo [22], Oliver Sakowitz [25, 26], Peter Smielewski [6], Nino Stocchetti [27], Nina Sundström [28], Riikka Takala [29], Tomas Tamosuitis [30], Olli Tenovuo [31], Andreas Unterberg [26], Peter Vajkoczy [32], Alessia Vargiolu [8], Rimantas Vilcinis [33], Stefan Wolf [34], Alexander Younsi [26], Frederick A. Zeiler [13,35]

[1] Department of Physical Medicine and Rehabilitation, University hospital Northern Norway

[2] Department of Neurology, Neurological Intensive Care Unit, Medical University of Innsbruck, Innsbruck, Austria

[3] Department of Neurosurgery & Anesthesia & intensive care medicine, Karolinska University Hospital, Stockholm, Sweden

[4] NeuroIntensive Care, Niguarda Hospital, Milan, Italy

[5] Department of Neurosurgery, Medical School, University of Pécs, Hungary and Neurotrauma Research Group, János Szentágothai Research Centre, University of Pécs, Hungary

[6] Brain Physics Lab, Division of Neurosurgery, Dept of Clinical Neurosciences, University of Cambridge, Addenbrooke's Hospital, Cambridge, UK

[7] Neuro ICU, Fondazione IRCCS Cà Granda Ospedale Maggiore Policlinico, Milan, Italy

[8] NeuroIntensive Care Unit, Department of Anesthesia & Intensive Care, ASST di Monza, Monza, Italy

[9] School of Medicine and Surgery, Università Milano Bicocca, Milano, Italy

[10] Department of Neurosurgery, Medical Faculty RWTH Aachen University, Aachen, Germany

[11] Department of Neurosurgery, University of Pecs and MTA-PTE Clinical Neuroscience MR Research Group and Janos Szentagothai Research Centre, University of Pecs, Hungarian Brain Research Program (Grant No. KTIA 13 NAP-A-II/8), Pecs, Hungary

[12] Department of Neurosurgery, University Hospitals Leuven, Leuven, Belgium

[13] Division of Anaesthesia, University of Cambridge, Addenbrooke's Hospital, Cambridge, UK

[14] Department of Anesthesiology and Intensive care, University Hospital Northern Norway, Tromso, Norway

[15] Neurointensive Care, Sheffield Teaching Hospitals NHS Foundation Trust, Sheffield, UK

[16] Departments of Neurology, Clinical Neurophysiology and Neuroanesthesiology, Region Hovedstaden Rigshospitalet, Copenhagen, Denmark

[17] Department of Clinical Neuroscience, Neurosurgery, Umeå University, Umeå, Sweden

[18] Department of Anaesthesiology, University Hospital of Aachen, Aachen, Germany

[19] Intensive Care Medicine, University Hospitals Leuven, Leuven, Belgium

[20] Department Neuroanesthesiology, Region Hovedstaden Rigshospitalet, Copenhagen, Denmark

[21] Helsinki University Central Hospital, Helsinki, Finland

[22] Department of Neurosurgery, Vall d'Hebron University Hospital, Barcelona, Spain

[23] Department of Neurosurgery, Kaunas University of technology and Vilnius University, Vilnius, Lithuania

[24] Department of Anaesthesia, Critical Care & Pain Medicine NHS Lothian & University of Edinburg, Edinburgh, UK

[25] Klinik für Neurochirurgie, Klinikum Ludwigsburg, Ludwigsburg, Germany

[26] Department of Neurosurgery, University Hospital Heidelberg, Heidelberg, Germany

[27] Department of Pathophysiology and Transplantation, Milan University, and Neuroscience ICU, Fondazione IRCCS Cà Granda Ospedale Maggiore Policlinico, Milano, Italy

[28] Department of Radiation Sciences, Biomedical Engineering, Umea University, Umea, Sweden

[29] Perioperative Services, Intensive Care Medicine, and Pain Management, Turku University Central Hospital and University of Turku, Turku, Finland

[30] Neuro-intensive Care Unit, Kaunas University of Health Sciences, Kaunas, Lithuania

[31] Rehabilitation and Brain Trauma, Turku University Central Hospital and University of Turku, Turku, Finland

[32] Neurologie, Neurochirurgie und Psychiatrie, Charité–Universitätsmedizin Berlin, Berlin, Germany

[33] Department of Neurosurgery, Kaunas University of Health Sciences, Kaunas, Lithuania

[34] Department of Neurosurgery, Charité–Universitätsmedizin Berlin, corporate member of Freie Universität Berlin, Humboldt-Universität zu Berlin, and Berlin Institute of Health, Berlin, Germany

[35] Section of Neurosurgery, Department of Surgery, Rady Faculty of Health Sciences, University of Manitoba, Winnipeg, MB, Canada

## Author Contributions

**Conceptualization:** Cecilia AI Åkerlund, Joseph Donnelly, Raimund Helbok, Anders Holst, Marek Czosnyka, Peter Smielewski, Nino Stocchetti, Ari Ercole, David W. Nelson.

**Data curation:** Cecilia AI Åkerlund, Frederick A. Zeiler, Manuel Cabeleira.

**Formal analysis:** Cecilia AI Åkerlund, Joseph Donnelly, Ari Ercole, David W. Nelson.

**Methodology:** Cecilia AI Åkerlund, Joseph Donnelly, Anders Holst, Fabian Güiza, Geert Meyfroidt, Ari Ercole, David W. Nelson.

**Software:** Cecilia AI Åkerlund, Joseph Donnelly, Marek Czosnyka, Peter Smielewski.

**Supervision:** David W. Nelson.

**Visualization:** Cecilia AI Åkerlund, Joseph Donnelly.

**Writing – original draft:** David W. Nelson.

**Writing – review & editing:** David W. Nelson.

**Writing – original draft:** Cecilia AI Åkerlund, Anders Holst, Ari Ercole.

**Writing – review & editing:** Cecilia AI Åkerlund, Joseph Donnelly, Frederick A. Zeiler, Raimund Helbok, Anders Holst, Manuel Cabeleira, Fabian Güiza, Geert Meyfroidt, Marek Czosnyka, Peter Smielewski, Nino Stocchetti, Ari Ercole.

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
