## [Decision Letter · Decision Letter 0]

21 Oct 2020

PONE-D-20-31465

Impact of duration and magnitude of raised intracranial pressure on outcome after severe traumatic brain injury: A CENTER-TBI High resolution group study

PLOS ONE

Dear Dr. Åkerlund,

Thank you for submitting your manuscript to PLOS ONE. After careful consideration, we feel that it has merit but does not fully meet PLOS ONE’s publication criteria as it currently stands. Therefore, we invite you to submit a revised version of the manuscript that addresses the points raised during the review process.

We look forward to receiving your revised manuscript.

Kind regards,

Firas H Kobeissy, PhD

Academic Editor

PLOS ONE

Journal Requirements:

2.Thank you for including your ethics statement:  "The CENTER-TBI study (EC grant 602150) has been conducted in accordance with all relevant laws of the EU if directly applicable or of direct effect and all relevant laws of the country where the Recruiting sites were located, including but not limited to, the relevant privacy and data protection laws and regulations (the “Privacy Law”), the relevant laws and regulations on the use of human materials, and all relevant guidance relating to clinical studies from time to time in force including, but not limited to, the ICH Harmonised Tripartite Guideline for Good Clinical Practice (CPMP/ICH/135/95) (“ICH GCP”) and the World Medical Association Declaration of Helsinki entitled “Ethical Principles for Medical Research Involving Human Subjects”. Informed Consent by the patients and/or the legal representative/next of kin was obtained, accordingly to the local legislations, for all patients recruited in the Core Dataset of CENTER-TBI and documented in the e-CRF.

Ethical approval was obtained for each recruiting site. The list of sites, Ethical Committees, approval numbers and approval dates can be found on the website: . " ext-link-type="uri" xlink:type="simple">https://www.center-tbi.eu/project/ethical-approval".   

3.We note that you have indicated that data from this study are available upon request. PLOS only allows data to be available upon request if there are legal or ethical restrictions on sharing data publicly. For information on unacceptable data access restrictions, please see http://journals.plos.org/plosone/s/data-availability#loc-unacceptable-data-access-restrictions.

4.Thank you for stating the following in the Financial Disclosure  section:

[Data used in preparation of this manuscript were obtained in the context of CENTER-TBI, a large collaborative project with the support of the European Union 7th Framework program (EC grant 602150). Additional funding was obtained from the Hannelore Kohl Stiftung (Germany), from OneMind (USA) and from Integra LifeSciences Corporation (USA). FAZ receives research support from the Manitoba Public Insurance (MPI) Neuroscience/TBI Research Endowment, the Health Sciences Centre Foundation Winnipeg, the United States National Institutes of Health (NIH) through the National Institute of Neurological Disorders and Stroke (NINDS), the Canadian Institutes for Health Research (CIHR), the Canadian Foundation for Innovation (CFI), the University of Manitoba Centre on Aging, the University of Manitoba VPRI Research Investment Fund (RIF), and the University of Manitoba Rudy Falk Clinician-Scientist Professorship. DN has also been funded by the Regional Research Agreement (ALF) with Stockholm City Council.

The funders had no role in study design, data collection and analysis, decision to publish, or preparation of the manuscript.]. 

We note that you received funding from a commercial source: [Integra LifeSciences Corporation]

5. One of the noted authors is a group or consortium [ENTER-TBI High Resolution ICU Sub-Study Participants and Investigators]. In addition to naming the author group, please list the individual authors and affiliations within this group in the acknowledgments section of your manuscript. Please also indicate clearly a lead author for this group along with a contact email address.

6. Please include a copy of Table 5 which you refer to in your text on page 15.

Additional Editor Comments (if provided):

Dear Drs Åkerlund and Nelson,

this is a very nice work that has been evaluated by top researchers in the area of TBI.

i would request adding some sections related to limitations, shortcomings and future directions.

Best

Reviewers' comments:

Reviewer's Responses to Questions

**Comments to the Author**

1. Is the manuscript technically sound, and do the data support the conclusions?

Reviewer #1: Yes

Reviewer #2: Yes

2. Has the statistical analysis been performed appropriately and rigorously? 

Reviewer #1: Yes

Reviewer #2: Yes

3. Have the authors made all data underlying the findings in their manuscript fully available?

Reviewer #1: Yes

Reviewer #2: Yes

4. Is the manuscript presented in an intelligible fashion and written in standard English?

Reviewer #1: Yes

Reviewer #2: Yes

5. Review Comments to the Author

Reviewer #1: minor revisions: show mmhgxhrs rather than using astrc * or use a centered dot.

Describe in text more detail about how autoregulation measures were assessed in simple terms

In discussion include a formal section on advantages and limitations...

Reviewer #2: In the present manuscript titled “Impact of duration and magnitude of raised intracranial pressure on outcome after severe traumatic brain injury: A CENTER-TBI High resolution group study” authors Dr. Åkerlund et al., presented a study to investigate the impact of intracranial pressure (ICP) intensity and duration on outcome in the large multi-center cohort in the CENTER-TBI study. The main finding is the relationship between time-dependent ICP insults and outcome. The authors identify clear relationship between high doses of ICP and mortality but not for favourable/unfavourable outcome. Also, clear correlations between number of events above thresholds of ICP intensity, although short durations and worse outcome

The bootstrap methodology to assess the certainty/uncertainty of the transition line of the correlation plot presents a novel and main contribution to the field of TBI.

The data showing relations of potential ICP vulnerability during periods of intact and impaired autoregulation and suggestions that safe ICP levels may vary depending on autoregulatory status, is a critical data that adds significance to the field of TBI.

The manuscript narrative is well written. The methodology and the data analysis is conducted thoroughly. The discussion and conclusions are appropriately conveyed. Finally the hypothesis that short periods of raised ICP may be causative of injury and long periods of high ICP may be associative with injury severity is relevantly acknowledged.

I recommend that this paper be accepted as is.

6. PLOS authors have the option to publish the peer review history of their article (what does this mean?). If published, this will include your full peer review and any attached files.

Reviewer #1: No

Reviewer #2: No

---

## [Author Response · Author response to Decision Letter 0]

18 Nov 2020

Dear Dr Kobeissy and the Reviewers,

Thank you for your encouraging words and quick reply, and for valuable comments. We have carefully considered all of them, and made changes to the manuscript accordingly, as can be seen in the attached manuscript files. Here follows replies to all your comments.

Thank you for stating this. We have carefully gone through the manuscript and updated it to adhere to the style requirements. Please let us know if the manuscript should be further updated. 

Please provide additional details regarding participant consent. In the ethics statement in the Methods and online submission information, please ensure that you have specified (1) whether consent was informed and (2) what type you obtained (for instance, written or verbal, and if verbal, how it was documented and witnessed). If your study included minors, state whether you obtained consent from parents or guardians. If the need for consent was waived by the ethics committee, please include this information.

- The ethics statement has been updated, in the methods section as well as in the online submission form, to include a reference to the ethical approvals for the sites in the high-resolution group, which recruited patients for this study. The type of consent has been clarified.

Modifications:

“This study was approved by the CENTER-TBI management committee. The CENTER-TBI study was conducted in accordance with all relevant laws of the European Union if directly applicable or of direct effect and all relevant laws of the country where the Recruiting sites were located, including but not limited to, the relevant privacy and data protection laws and regulations (the “Privacy Law”), the relevant laws and regulations on the use of human materials, and all relevant guidance relating to clinical studies from time to time in force including, but not limited to, the ICH Harmonised Tripartite Guideline for Good Clinical Practice (CPMP/ICH/135/95) (“ICH GCP”) and the World Medical Association Declaration of Helsinki. Written or oral Informed Consent by the patients or next of kin was obtained, according to the local legislations, for all patients recruited in the Core Dataset of CENTER-TBI and documented in the electronic case report form. In case of oral consent, a written confirmation was requested.

Ethical approval was obtained for each recruiting site. The list of sites, Ethical Committees, approval numbers and approval dates are available online [21] and ethical approval numbers for sites having recruited patients to the high-resolution sub-study of CENTER-TBI is listed in S1 Appendix.”

- It is not possible at this time point to share the data set freely due to the complexity of the data and a clear risk of identifying individual patients. However, if a legitimate scientific interest or need exists, it is possible to contact the CENTER-TBI steering committee for permission. Therefore, we have updated the data sharing paragraph to the following:

“CENTER-TBI encourages data sharing, and there is a data sharing statement published: https://center-tbi.eu/data/sharing. Ethical and potential legal restrictions apply due to concerns regarding potential accidental re-identification of patients from such a complex and comprehensive dataset. Data will be available to researchers who provide a study proposal that is approved by the management committee to achieve the aims in the approved proposal. Proposals can be submitted online at https://www.center-tbi.eu/data. A request to access data to control analysis of a published article will be addressed in the same manner. A data access agreement is required and all access must comply with regulatory restrictions imposed on the original study.”

4. We note that you received funding from a commercial source: [Integra LifeSciences Corporation]. Please provide an amended Competing Interests Statement that explicitly states this commercial funder, along with any other relevant declarations relating to employment, consultancy, patents, products in development, marketed products, etc. 

- Thank you for mentioning this. We would like to clarify Integra LifeSciences role in the study by changing the Competing Interests Statement as stated in the Revision Cover Letter.

5. One of the noted authors is a group or consortium [ENTER-TBI High Resolution ICU Sub-Study Participants and Investigators]. In addition to naming the author group, please list the individual authors and affiliations within this group in the acknowledgments section of your manuscript. Please also indicate clearly a lead author for this group along with a contact email address.

- We have added the High resolution sub-study participants and investigators to the acknowledgement.

6. Please include a copy of Table 5 which you refer to in your text on page 15.

- Thank you for finding this typo, the reference has been corrected to table 4. 

Additional Editor Comments (if provided):

Dear Drs Åkerlund and Nelson,

this is a very nice work that has been evaluated by top researchers in the area of TBI.

i would request adding some sections related to limitations, shortcomings and future directions.

- Thank you for your kind words, happy to hear that you appreciate our work. As requested, a section concerning limitations, advantages and future directions have been added to the discussion.

Reviewers comments

Reviewer #1: 

1. show mmhg x hrs rather than using astrc * or use a centered dot.

- Thank you for noting this. Asterices have been changed to centered dots. 

2. Describe in text more detail about how autoregulation measures were assessed in simple terms

- We have also added a short description on how autoregulation measurements were calculated and assessed, in the method section.

3. In discussion include a formal section on advantages and limitations...

- In line with requests from reviewer #1 and Editor Dr Kobeissy, we have added a sub-section in the discussion section on advantages, limitations and future directions.

Once again, thank you for a quick and encouraging reply. We hope that you agree that our updates have improved the manuscript. Looking forward to your feedback soon.

Best wishes,

Cecilia Åkerlund and David Nelson

---

## [Decision Letter · Decision Letter 1]

23 Nov 2020

Impact of duration and magnitude of raised intracranial pressure on outcome after severe traumatic brain injury: A CENTER-TBI High resolution group study

PONE-D-20-31465R1

Dear Dr. Åkerlund,

We’re pleased to inform you that your manuscript has been judged scientifically suitable for publication and will be formally accepted for publication once it meets all outstanding technical requirements.

Kind regards,

Firas H Kobeissy, PhD

Academic Editor

PLOS ONE

Additional Editor Comments (optional):

Reviewers' comments:

Reviewer's Responses to Questions

**Comments to the Author**

1. If the authors have adequately addressed your comments raised in a previous round of review and you feel that this manuscript is now acceptable for publication, you may indicate that here to bypass the “Comments to the Author” section, enter your conflict of interest statement in the “Confidential to Editor” section, and submit your "Accept" recommendation.

Reviewer #1: All comments have been addressed

Reviewer #2: All comments have been addressed

2. Is the manuscript technically sound, and do the data support the conclusions?

Reviewer #1: Yes

Reviewer #2: Yes

3. Has the statistical analysis been performed appropriately and rigorously? 

Reviewer #1: Yes

Reviewer #2: Yes

4. Have the authors made all data underlying the findings in their manuscript fully available?

Reviewer #1: Yes

Reviewer #2: Yes

5. Is the manuscript presented in an intelligible fashion and written in standard English?

Reviewer #1: Yes

Reviewer #2: Yes

6. Review Comments to the Author

Reviewer #1: Thank you for the responses to comments. The report has been improved and promises insights into the management of TBI based on ICP measurements.

Reviewer #2: The authors have addressed comments raised previously. This manuscript is now acceptable for publication in the current format.

7. PLOS authors have the option to publish the peer review history of their article (what does this mean?). If published, this will include your full peer review and any attached files.

Reviewer #1: **Yes: **Ralph G DePalma MD

Reviewer #2: No

---

## [Editor Report · Acceptance letter]

1 Dec 2020

PONE-D-20-31465R1 

Impact of duration and magnitude of raised intracranial pressure on outcome after severe traumatic brain injury: A CENTER-TBI high-resolution group study 

Dear Dr. Åkerlund:

I'm pleased to inform you that your manuscript has been deemed suitable for publication in PLOS ONE. Congratulations! Your manuscript is now with our production department. 

Kind regards, 

on behalf of

Dr. Firas H Kobeissy 

Academic Editor

PLOS ONE